# Monolithic 45 Degree Deflecting Mirror as a Key Element for Realization of 2D Arrays of Laser Diodes Based on AlInGaN Semiconductors

**DOI:** 10.3390/mi14020352

**Published:** 2023-01-31

**Authors:** Kiran Saba, Anna Kafar, Jacek Kacperski, Krzysztof Gibasiewicz, Dario Schiavon, Takao Oto, Szymon Grzanka, Piotr Perlin

**Affiliations:** 1Institute of High Pressure Physics, Polish Academy of Sciences, 01-142 Warsaw, Poland; 2TopGaN Limited, 01-142 Warsaw, Poland; 3Department of Informatics and Electronics, Yamagata University, Yamagata 992-8510, Japan

**Keywords:** 2D-arrays, AlInGaN, semiconductor lasers, edge-emitting lasers, integrated mirrors

## Abstract

In this study, we propose a solution for realization of surface emitting, 2D array of visible light laser diodes based on AlInGaN semiconductors. The presented system consists of a horizontal cavity lasing section adjoined with beam deflecting section in the form of 45° inclined planes. They are placed in the close vicinity of etched vertical cavity mirrors that are fabricated by Reactive Ion Beam Etching. The principle of operation of this device is confirmed experimentally; however, we observed an unexpected angular distribution of reflected rays for the angles lower than 45°, which we associate with the light diffraction and interference between the vertical and deflecting mirrors. The presented solution offers the maturity of edge-emitting laser technology combined with versatility of surface-emitting lasers, including on-wafer testing of emitters and addressability of single light sources.

## 1. Introduction

InGaN/GaN-based visible light laser diodes, due to their high brightness and high efficiency [1], are the subject of many important applications in solid-state lighting [2,3], display applications [2,4], car headlights [5,6], projectors [7,8], atomic clocks [9,10], and laser-based high-density storage systems [11]. Moreover, these devices are crucial components for emerging visible light communication systems [12,13,14]. All applications that utilize visible laser diode technologies are based so far on edge-emitting devices (Edge-Emitting Lasers—EEL). This technology offers high optical power, good beam, and spectral quality, along with controlled lifetime and general maturity. However, edge-emitting nitride semiconductor-based laser diodes have several important drawbacks, including the lack of possibility of on-wafer testing [13], complicated and costly processing, packaging, difficulty in the production of laser diode bars (1D solution), and stacked laser diodes (2D solution). The two latter issues are related to thermal challenges and wafer bow characteristics for highly strained InGaAlN laser diode heterostructures. Fabrication of 1D and especially 2D arrays could help realize not only compact, high-power emitters needed for Digital Light Processing (DLP) projectors [7,8] and automobile headlights [5,6] but also lower-power systems such as addressable arrays of laser diode for fast direct visual telecommunication. The goals presented above could be in part achieved by using arrays of Vertical Cavity Surface-Emitting Lasers (VCSELs) [15,16]. These devices have light emission perpendicular to the plane of the chip [17] and circular beam cross section facilitating the light coupling to optical fibers [18,19]. In addition, testing such individual laser devices does not require wafer dicing or mounting and can be tested on wafer before packaging [20,21]. However, nitride-semiconductor VCSELs are not yet mature [22] and need to be developed to the industrial level, along with the fact that they are low-power sources in comparison with EELs [23].

In this work, with the purpose of mitigating the existing limitations of both EELs and VCSELs, we present a Surface Emitting Laser diode (SEL) with a horizontal lasing cavity and additional 45° etched mirrors for beam deflection and surface emission. As light is emitted from the surface of the device, this not only allows easy power scaling but also results in better thermal properties and less wafer bowing, along with easy coupling of emitted light beam with other optoelectronic elements. The proposed 2D array emitters can be individually addressed and controlled, which enables homogenous operation of the array as a whole. In case of stacked laser bars, they are usually cooled by having micro-channel cooling plates in–between the bars. Our design provides a 2D array configuration with a possibility of cooling by simply placing the array on a standard cooling plate. In addition, the distance of 900 µm used between the emitters prevents them from thermal crosstalk. Regarding the wafer bowing problem, our wafer does not need to be thinned down and can stay much thicker than cleaved laser diode bars. We expect to reduce the bow of the laser diode structure by a factor of 10 (Stoney Equation [24]). Regarding the packaging, the stacked bars do not have any standard packaging, while our 2D arrays can be mounted, for example, in the standard QFN packages.

This research idea was previously utilized by J. P. Donnelly et al., for the fabrication of monolithic 2D arrays based on GaAs/AlGaAs laser diodes, whereas vertical laser facets and deflecting mirrors were fabricated by ion beam-assisted etching [25]. Likewise, N. Hamao et al., reported SEL diodes based on GaAs/AlGaAs with 45° total reflection micro mirrors formed by Reactive Ion Etching (RIE) method within 1° of precision of 45° angle [26].

Moreover, other research studies have shown monolithically mirror-integrated surface-emitting laser (SEL) diodes on GaN, silicon, as well as on InP platforms. In 2008, Masao Kawaguchi et al., reported a novel approach that experimentally fabricates GaN-based surface-emitting laser diode, with a horizontal laser cavity having a 45° inclined mirror formed by Focus Ion Beam (FIB) etching for GaN-based 2D laser array applications. Fabricated LDs lased at the wavelength of 390 nm with the threshold current of 260 mA and the beam divergence angle of 24°, perpendicular to the junction plane [27]. Recently, Dominika Dąbrówka et al., presented a computational thermal analysis of surface-emitting 2D laser diode arrays on a GaN platform. The investigated structure consisted of 8 × 8 emitters with horizontal lasing section combined with 45° deflecting mirrors. In addition to showing the self-heating effect of an individual emitter on the GaN platform, the authors provided a comparison of temperature increase between GaN and hypothetical GaAs substrates [28].

Furthermore, in 1991, B. Stegmuller et al., reported surface-emitting distributed feedback laser diodes based on InGaAsP/InP integrated with a 45° deflecting mirror, which was fabricated by wet-etching method. However, beam collimation was achieved by ion beam-etched micro-lenses [29].

Suzuki et al., reported 1.3 µm lens-integrated SEL (based on an InP distributed feedback laser) along with a grating coupler, and both are fabricated on silicon-on-insulator (SOI) substrate [30]. In 2012, Inoue et al., proposed integration of embedded 45° mirrors tapered in a dielectric glass planer waveguide (trench was made by dry-etching) by employing the liquid immersion exposure technique [31]. In 2013, Chin-Ta Chen et al., proposed the polymer waveguide with silicon 45° terminated micro-reflectors fabricated by chemical wet etching for on-chip and out-of-plane coupling [32]. Recently, in 2020, Akihiro Noriki et al., utilized the 45° curved micro-mirror for broadband silicon (Si) photonics vertical coupling. Curved micro-mirrors were integrated on Si-photonics by grey-scale photolithography with laser–writer [33]. In addition, with easy power scaling, easy coupling, and cost reduction, utilization of GaN platforms for the fabrication of proposed 2D arrays will have better thermal properties and device structure, owing to the very high thermal conductivity of GaN.

## 2. Device Design

Fully processed laser epi-wafer served as a starting point of the fabrication process described below. The approach to construct fully monolithic InGaN-based laser diode with the beam direction changed from horizontal to vertical is not quite trivial. The experimental design must meet at least three basic requirements:The angle of inclination of the mirrors should be close to 45°.The deflecting mirrors should be very closely located with relation to vertical facet of the lasers to allow all of the emitted beam to be reflected.The whole structure should be reasonably easy to process.

First, the second point mentioned above is analyzed. Figure 1a. shows the most obvious configuration of such a device. As the laser diode waveguide is located around 0.7 μm below the surface of the structure and typical beam divergence in the fast axis is around 30°, we observe that part of the light beam is not reflected by the processed mirror, and basically only a lower part of the beam can be vertically outcoupled. Ideally, there should not be any distance between both vertical laser cavity mirror and 45° deflecting mirror; however, owing to the complexity of the alignment of consecutive device processing steps, it is beneficial to leave distance between the two. As a result, the photolithography mask is designed to have 0.5–1 µm distance between both features, as this will improve the chances of better alignment of the waveguide with the center of the 45° mirror.

The improved geometry of the structure is shown in Figure 1b. Note the position of the active layer of the laser coincides with the 45° mirror center. The necessity of aligning the deflecting mirror and the emitting layer of the laser excludes using completely planar structure. Figure 2 shows the whole concept of the experimental device design, containing two dry-etched vertical mirrors defining the laser cavity and two deflecting mirrors. Below, we explain the fabrication of such a structure.

## 3. Fabrication Scheme

The fabrication scheme of the proposed experimental device design consists of the following steps as shown in Figure 3.

Steps 1 and 2 of the fabrication scheme (Figure 3) have the greatest importance for the functionality of the presented device. These steps are not only for the fabrication of deflecting 45° mirrors (in the initial position) but also allowingfor locating the active area of the laser at the base of the deflecting mirrors. This allows for subsequent positioning of the deflecting mirrors with respect to the active region during step 6 dry etching.

## 4. Methodology

The methodology of first two steps are described first (Figure 3). As a first step of device fabrication, free standing (0001) GaN substrates were patterned through optimized binary photolithography followed by transferring of the pattern with dry etching (Ar and Cl ions-Oxford PlasmaLab100). For the origination of 45° slopes experimentally through binary photolithography, we considered photoresist exposure with mask aligner system within a proximity mode, which is also known as “shadow printing” (Figure 4) [34,35]. The proximity mode of mask-aligner technique permits the exposure of photoresist having a significant separation distance from the surface of the used photomask. To find the best proximity distance, we performed many experiments where we studied multiple distances between the mask and the sample: 20–100 µm and changed the dose for exposition along with the time of development. After making each set of photolithography experiments, we studied the slopes by a stylus profilometer (both the shape of the slope and the angle of inclination). In our systematic study, we fixed the optimal distance in the range 40–60 μm for light dose of 35 mJ/cm^2^ and development time of 2 min.

## 5. Device Fabrication and Processing

For experimental fabrication of 45° inclined mirrors, we performed multilevel photolithography (Figure 3, step 1). Multilevel photolithography (also called grey scale photolithography) is based on illuminating a positive photoresist with spatially varying light doses. This is usually carried out by using a laser writing machine where it is possible to change the intensity of the laser between neighboring positions while scanning. Normally, a positive photoresist is bleached by the light through its whole thickness. When a thick photoresist is used and the light doses are chosen precisely, it is possible to bleach the resist layer only up to, e.g., half of the total thickness. During the development, this leads to washing away half of the resist thickness. By properly choosing the special change in light doses, it is possible to obtain spatial variation of the thicknesses after the development, opening a possibility to create 3D structures from the resist. However, in this work instead of the use of laser-controlled dose, we used the shadow lithography with a mask-aligner system. This approach is technologically better as it is faster and more stable.

In Figure 3, step 1 is performed by utilizing a positive photoresist Az9260 (6 µm) spin coated on the sample. Followed by the pre-bake of 100 °C for 1 min and thereafter, ultraviolet exposure (without the mask) with a constant 6.9 mJ/cm^2^ dose for 1 s, which is regarded as the background level illumination in our optimized binary photolithography parameters. Background level illumination is needed to utilize for patterning the full thickness of the resist after development. Afterwards, sample was exposed through quartz mask for another 6 s. The separation between the mask and the surface of the photoresist allows the free space propagation of light wave, which leads to the light diffraction. This results in a non-abrupt exposure of the photoresist at the edge of the pattern of the mask (Figure 4). Around the edge, the light intensity varies gradually from zero to full intensity, which produces sloped walls after the development step (Figure 4). The pattern was then transferred onto the substrate by dry etching with Ar/Cl_2_ plasma (24 sccm & 8 sccm, respectively,) in an Inductively Coupled Plasma Reactive Ion Etching (ICP RIE) system with RF 40 W, ICP 250 W and pressure of 10 mT.

After transferring of the desired photolithography pattern onto the substrate with dry etching, epitaxial layers were grown on the patterned substrate by Metalorganic Vapor Phase Epitaxy (MOVPE) with a set of two staggered quantum wells. The epitaxial structure of the device under study with details of thickness and composition of the grown layers is described below in Figure 5.

The height of the ridge is 810 nm. Further details of the typical device design and operating parameters of our edge emitting laser diodes could be found in the previous reported research [36].

After the epitaxy, we performed a standard processing scheme including electrical metal contact deposition, etching of the ridge waveguide, depositing electrically insulating SiO_2_ layer, contact annealing, etc. (step 4 in Figure 3). We defined the laser cavity stripes to be 900 μm long and the laser ridge to be 2 μm wide. We also performed additional processing steps that allowed us to define the vertical mirrors of the edge-emitting laser and at the same time to shift the 45° slopes to their final position facing the output of the laser waveguide (Figure 3, steps 5 and 6). This photolithography step was performed using a negative photoresist to ensure better vertical orientation and smoothness of the vertical laser mirror. The depth of the etching was calculated based on the position of the active region of the laser with respect to the top of the laser area and the height of the 45° slope with the final goal of lowering the 45° slopes in such a way that the middle of its height is at the same height as the quantum wells of the laser. As the last step, we used tetramethylammonium hydroxide (TMAH) solution (27%), which was maintained at a constant temperature of 80 °C and stirred continuously for 20 min in order to improve the smoothness of the vertical mirror. The etching by TMAH is crystallographically selective. It reveals the most stable plane-m-plane (1–100). It is explained by the attachment of hydroxide ions to the positively charged Ga dangling bonds; however, access to the Ga atoms is restricted by the neighboring negatively charged N dangling bonds. It has been reported earlier as well that under similar conditions, etching is usually slow for m-plane and normally does not etch the c-plane. Slowly etched m-planes will result in vertical laser facets, as presented in a previous research study [37].

## 6. Experimental Results and Discussion

For the sake of slope angle confirmation experimentally, we performed the photolithography and dry-etching procedures on test samples with sapphire overgrown with 6 and 8 µm of GaN. One such sample with etched stripe patterns with inclined walls is shown in Figure 6. Figure 6a displays an SEM image of cleaved sample with an etched stripe representing a sloped edge. The confirmation of the angle at this aforesaid edge was carried out through stylus profilometer scans (Figure 6b).

Moreover, SEM images of the fully processed 2D arrays are presented in Figure 7a,b. Figure 7c shows the SEM images of the dry- and wet-etched facets of an individual proposed device fully possessed. It can be seen from the pictures that vertical laser facets are of very good quality [38].

For electrical and optical characterization (Figure 8), the array of laser diodes was mounted on top of copper heatsink, which provides relatively large heat capacity. The studies were performed under continuous wave (CW) operation and without temperature stabilization. Figure 8a lists the number of 8 × 8 emitters, whereas the red dotted lines indicate the number of devices for which LIV measurements are presented in Figure 8b,c. Preliminary results concluded that the device efficiency of a single horizontal-to-vertical emitter is as high as 0.6 W/A, which usually reaches up to 0.2 W/A/facet or above for the uncoated facets and a threshold current around 100 mA (Figure 8c).

The obtained threshold current is around 25% higher than that of the devices fabricated with analogous geometry, dimensions, and cleaved facet. We consider this a very good result, taking into account that the cavity mirrors were etched. The measured slope efficiency was seriously underestimated, firstly due to the direction of beam emission (this will be discussed in the next paragraph) and secondly due to the low reflectivity of the 45° slope at this stage as it was not covered by a high-reflective coating, which we will apply in our experiments in the future. One of the device number 8 in Figure 8b,c shows a higher threshold current and slope efficiency, and this is probably related to a difference in the shape (smoothness) of the vertical mirror in this device.

Figure 9 shows the light emission from an array of 4 × 2 emitters. All the emitters have similar device structure as presented in the section of device fabrication and processing. Fabricated devices are 1D TE polarized (transverse) having a fundamental single lasing mode and emitting between 430–435 nm.

Owing to the technical challenges related to the mounting of such a large 2D array, open copper mounts were utilized. The emitters were directly connected to the power meter, and the maximum emitted power was collected by the detector. This allowed us to perform electrical characterization and far-field measurements. As shown in Figure 9, there appears to be more emission from one side (from the top) and relatively less emission from the other facet. This could be attributed to the position of camera tilted more towards the right side than the left. 

The light emission from the developed devices had an untypical pattern resulting from the used geometry. Figure 10 shows the direction of emitting beam from an individual emitter taken from the 2D array of 4 × 2 emitters (Figure 9).

Below the lasing threshold when the spontaneous emission is dominating, we observe the reflected emission from the quantum wells as long stripes as shown in Figure 10a. Once the stimulated emission starts to dominate, a waveguide mode is formed and two bright spots are dominating with each corresponding to one end of the laser cavity (one redirected beam) as shown in Figure 10b, when the screen is placed on top. Those beams are also presented in Figure 10c with the screen placed in the plane containing the beams and laser waveguide axis. Unexpectedly, the beams show an inclined direction of emission and several sub-beams were observed.

To obtain the vertical emission from the proposed device, it is important to fabricate the deflecting 45° mirror with acceptable precision of its angular orientation (Figure 11). For practical applications, the use of the short focal length microlenses (e.g., Edmund Optics Fly’s Eye Array with focal length 0.6 mm and pitch of 300 μm) ensures good light collection even for mirrors inclination angle smaller or bigger than 45°. For 5° deviation from the designed 45° orientation, we expect the output angles in the range of +/− 10° from the vertical direction, which should be tolerable by microlenses.

One can see from Figure 11b that in order to keep the center of the output beam within +/− 10° from the vertical direction, the orientation of the deflecting mirror has to be retained between 40° and 50°. The experimental result presented in Figure 10c showing an oblique emission of the main beams suggests that the orientation of the deflecting mirror was different from the intended value.

To experimentally estimate the angle of the inclined mirror of the individual emitter, optical profilometer scans were carried out. One such estimated angle of 29°of the inclined mirror is presented in Figure 12.

A significant decrease in the angle of the mirror before and after full device processing can be easily visualized (Figure 6 and Figure 12). This can be attributed to the evolution of the shape of the inclined slope during the epitaxial growth of the device structure. This issue will be studied in the future experiments. A possible solution lies in increasing the angle of the deflecting mirror before epitaxial growth to compensate the changes of the shape during growth.

In Figure 13, intensity distribution of emitted light is measured with a far-field goniometric system, which is controlled by LabVIEW program, and it is connected to data acquisition module. Photodiode (scanner) is placed ~15 cm above the emitter, and it can scan profiles along a surface of a sphere with 15 cm diameter. Figure 13a shows experimentally measured far-field profile of a surface-emitting laser measured within the vertical plane including the two beam sources. The injection current was set to 90 mA above the laser diode threshold. The profile shows two nearly-Gaussian peaks (from both deflector mirrors of an individual emitter) with the highest intensity of around 24°. Figure 13b presents the intensity map measured in full angle. It shows two brighter spots corresponding to the two deflected beams. The presence of two beam sources in Figure 13a,b clearly indicates the two peaks emission from both 45° deflecting mirrors.

## 7. Simulation Results

Moreover, to investigate the reflected beam characterization theoretically in more details, finite-difference time-domain (FDTD) simulations were performed using commercial software (Poynting for Optics, Fujitsu Ltd, Tokyo, Japan). In the simulations, an InGaN/GaN waveguide with 2 µm width and a GaN-based mirror with an angle of β were prepared in the computational domain where each boundary is a perfect matching layer. The simple waveguide structure composed of only the InGaN waveguide and GaN cladding layers is assumed. Notably, it is difficult to take the QW structure into account because it takes much longer to calculate the properties due to the extremely small mesh size. The plane-wave light source (CW, λ = 430 nm, and TE-polarization) is set in the waveguide layer. The emitted light from the waveguide is reflected by the mirror and then changes the direction. To simulate the far-field radiation pattern of the reflected beam, an equivalent surface current (EQSC) at the plane above the mirror is calculated.

Figure 14a,b shows the radiation pattern. For reference, in Figure 14c, the data presented in Figure 13a is redrawn using similar style as in simulation results. Figure 14d,e shows electric field distribution at a certain time and for the cases of β = 45°and 30°, respectively. Note that the same values of mirror length and distance between the waveguide edge and mirror, *l* and *d*, is designed for the β = 45° and 30° cases. The interference between the direct and reflected beams occurs as Lloyd’s mirror. If the optical path difference is equal to *m*λ, where *m* is an integer, destructive interference occurs. As a result, several beam branches are observed, which is in accordance to what was observed experimentally in Figure 10c. Note that the experimentally observed interference was not much stronger because the beam interference was weakened by the light scattering due to the surface roughness of the mirror.

## 8. Results and Discussion

From the experimental results, it is evident that the proposed laser diodes emit light at the angles different from the intended vertical direction. Instead of obtaining a 45° deflecting mirror angle, a value of 29° was measured with respect to the horizontal direction. This is a consequence of the evolution of the orientation of the deflecting mirror during the epitaxial growth of the device structure and several device processing steps. However, device efficiency of the individual emitter with uncoated facets is up to 0.6 W/A, which is promising to obtain over 1 W/A efficiency for the structures with coated deflectors. For this class of lasers with cleaved mirrors, the threshold currents were around 50 mA, thus the results from this study-50–70mA-are satisfactory. The coating of the deflectors should lead to a very substantial increase of the optical power, at least a factor of two. The unexpected beam distribution further reduces the light coupling possibility. In the simulation results, the waveguide has the similar parameters as experimentally fabricated device structure. Simulations were performed for both 45°and 30° deflecting mirrors having a distance of 1 µm between the vertical laser facet and the deflecting mirror. As a consequence of destructive interference, multiple branches of the deflected beam were observed, similar to the experimental results in Figure 10c. Experimental results suggest that the main reason for the discrepancy between the experimentally observed shape of the output beam and the calculated one is the presence of roughness or imperfections on the deflector fabricated experimentally.

Nevertheless, considering the future devices, maintaining the right orientation of the deflecting mirrors throughout the whole fabrication process would be a primary objective. Concerning the simulation results, to achieve a better beam quality, the interference of light (producing additional side light beams) should be suppressed by, e.g., a narrower beam divergence.

Taken together, in this work, we demonstrated the required functionalities including surface outcoupling of light, addressability of individual emitters, and reasonable optoelectrical parameters.

## 9. Conclusions

We have demonstrated InGaN laser diodes on GaN substrate with monolithically integrated 45° mirrors, serving as deflectors to redirect light beams from horizontal to vertical directions. This opens the possibility of making 2D arrays of edge-emitting lasers acting as surface emitters, competitive with VCSELs (vertical Cavity Surface-Emitting Lasers). Our results point out at certain not yet completely solved problems, such as the exact control of the mirror angle, but also unexpected properties of a microcavity formed between vertical and inclined mirrors. The proximity of this two mirrors causes interference of light between micro-sized deflecting mirror and edge-emitting cavity mirrors and changes in beam propagation because of reflection at deflector surface. Moreover, Fresnel reflectivity of the deflector is nearly 50%, and partially incident beam may be lost due to coupling into the deflectors. However, potential solutions, such as changing the shape of the deflector along with tilting angle and employing HR coatings, could hinder the unpredicted beam propagation after deflection.

Lastly, we conclude that fabrication of the proposed devices on high quality patterned GaN substrates have optoelectronic parameters comparable with the standard cleaved lasers fabricated on the same type of epitaxial structure.

## Figures and Tables

**Figure 1 micromachines-14-00352-f001:**
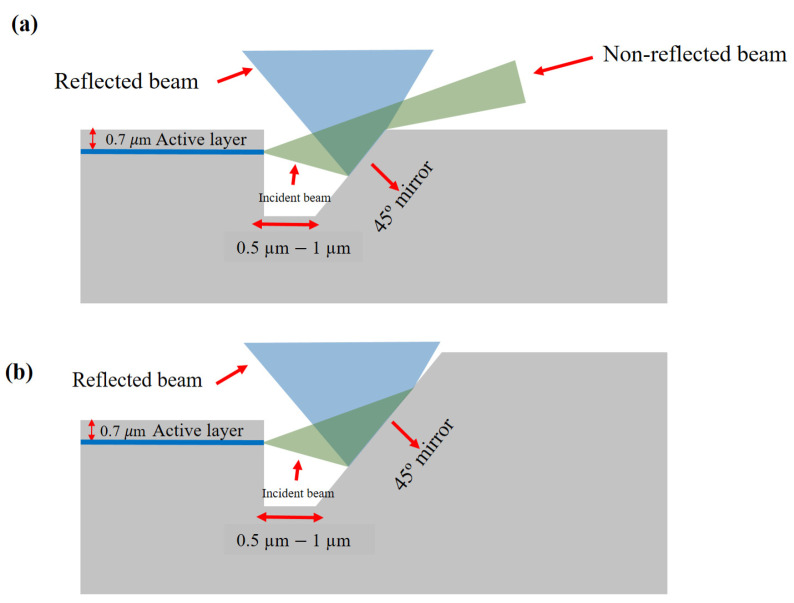
The simplest geometry of the laser structure with etched vertical laser mirror and deflective 45° mirror (**a**) device made on planar structure (**b**) laser fabricated below the surface of the wafer.

**Figure 2 micromachines-14-00352-f002:**
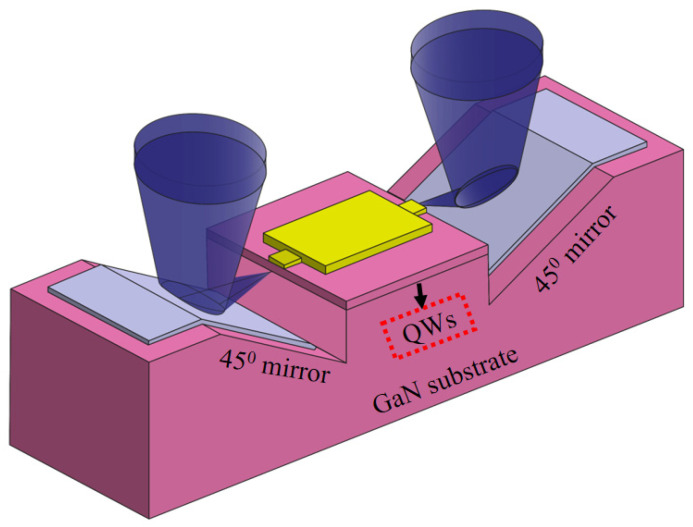
Cross sectional view of an individual emitter with the horizontal laser cavity adjoined with beam defecting 45° mirrors, whereas the red dotted lines indicate the position of the active region with Quantum Wells (QWs).

**Figure 3 micromachines-14-00352-f003:**
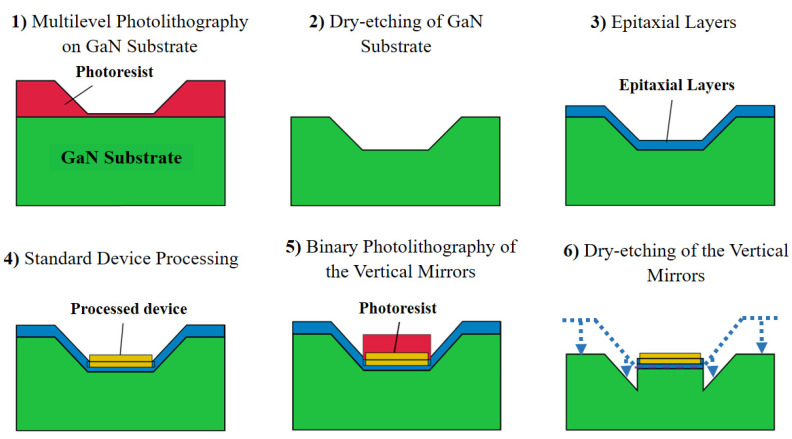
Scheme of consecutive device processing steps from GaN patterning to formation of vertical and deflecting mirrors.

**Figure 4 micromachines-14-00352-f004:**
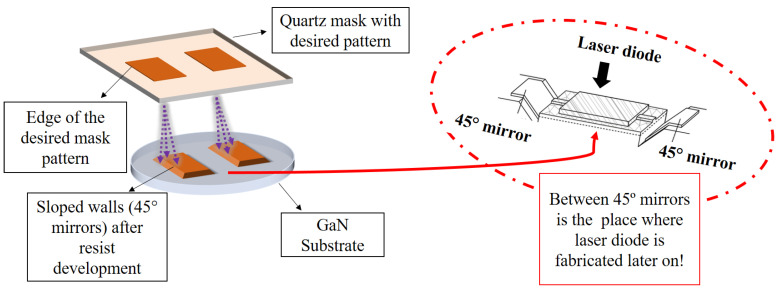
Photolithography for patterning of GaN substrate through mask-aligner system. Quartz mask indicates the edge of the desired pattern at which shadow appears during the exposure and spreads when the distance between the mask and the surface varies accordingly. The distance between the photomask and wafer is retained at 40–60 μm.

**Figure 5 micromachines-14-00352-f005:**
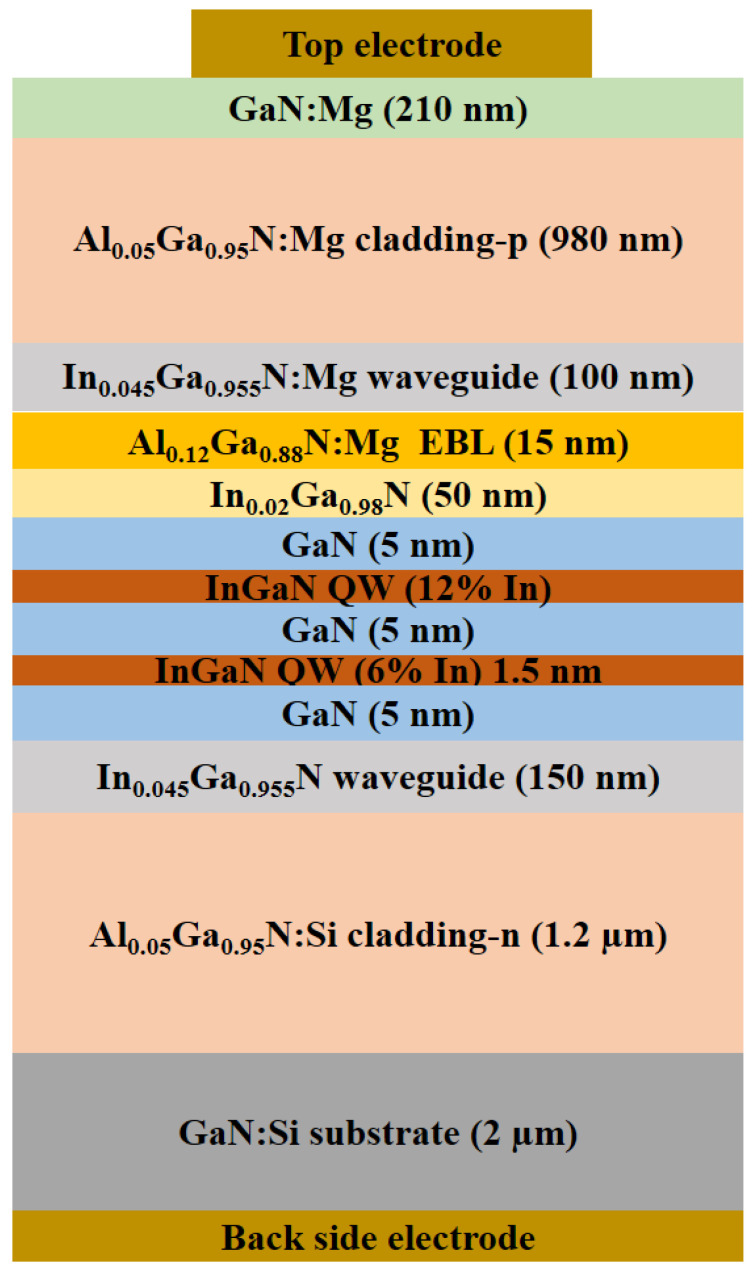
Cross-sectional view of the epitaxial structure of the proposed device design.

**Figure 6 micromachines-14-00352-f006:**
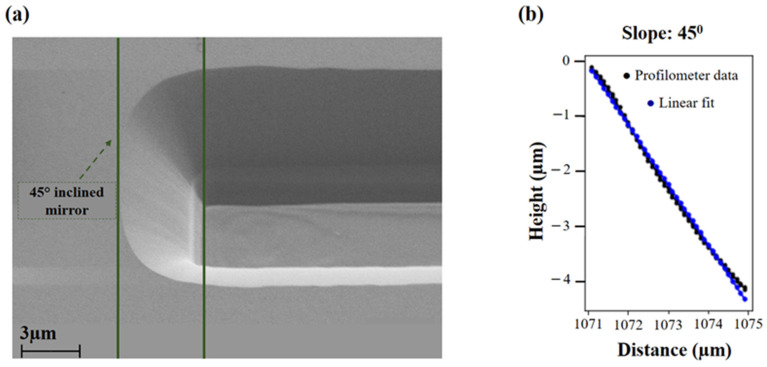
(**a**) Scanning Electron Microscope (SEM) image of a sample with etched 45° slopes fabricated by mask-aligner technology. (**b**) Stylus profilometer scan of the edge confirming the slope angle of figure (**a**).

**Figure 7 micromachines-14-00352-f007:**
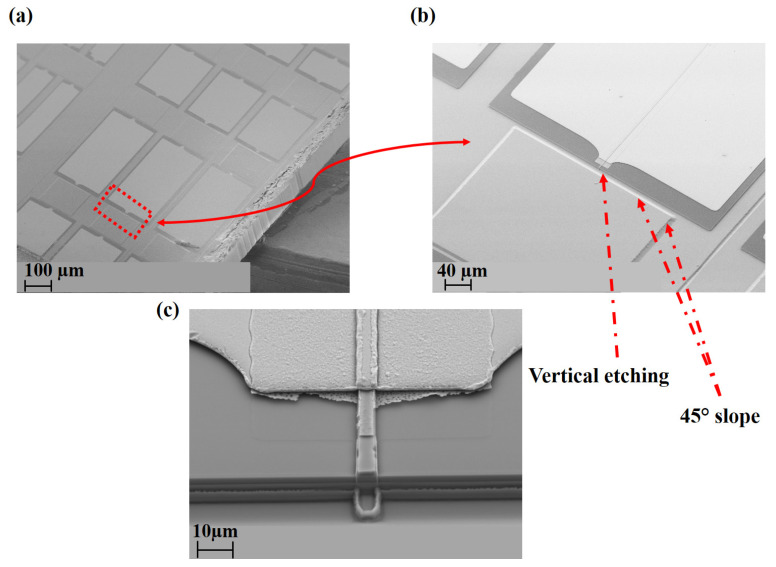
(**a**) SEM images of the fully processed 2D arrays (**b**) enlarged SEM image of an individual emitter with 45° inclined planes (**c**) zoomed in SEM image of the etched vertical laser facet of an individual emitter with horizontal lasing cavity from the top.

**Figure 8 micromachines-14-00352-f008:**
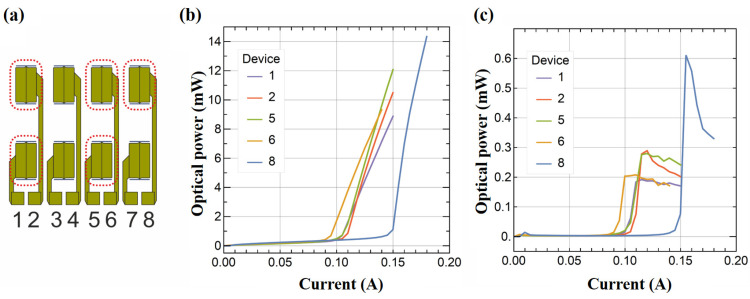
(**a**) 2D array of 4 × 2 emitters, (**b**) light vs. current dependences obtained for a series of edge-emitting devices equipped with the 45° deflectors, and (**c**) corresponding slope efficiency as a dependence on current.

**Figure 9 micromachines-14-00352-f009:**
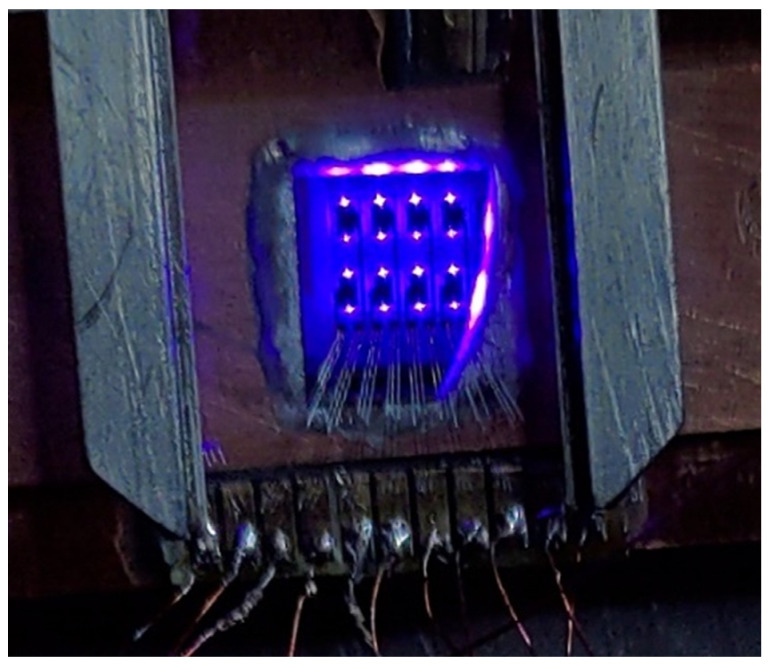
Light emission from the proposed 2D array of horizontal-to-vertical surface- emitting laser diodes.

**Figure 10 micromachines-14-00352-f010:**
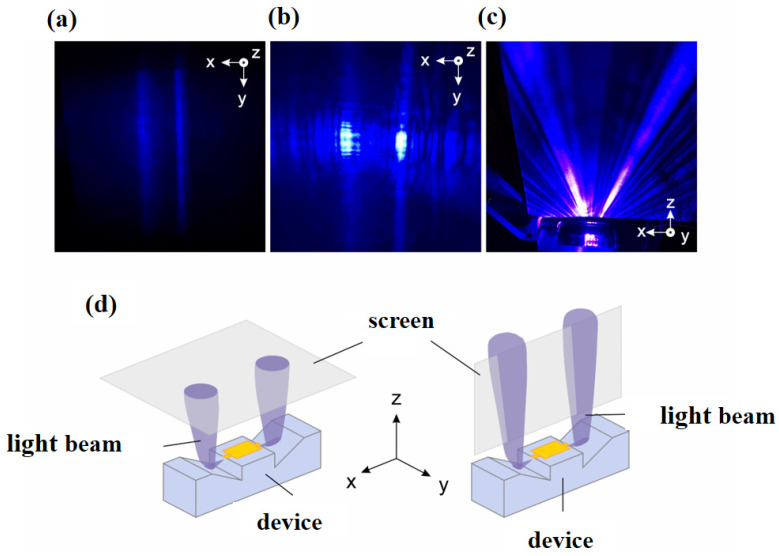
Photographic images show the direction of beam emission, (**a**) below laser threshold, (**b**) above laser threshold, (**c**) when the screen is placed in the plane containing the beams and laser waveguide axis, and (**d**) schematic view of how the emitted beams direction was captured in figure (**a**–**c**).

**Figure 11 micromachines-14-00352-f011:**
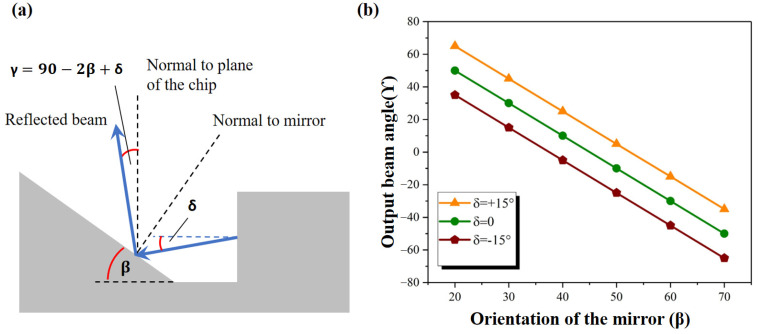
(**a**,**b**). Scheme of how theoretically the output beam angles are computed with respect to the various orientations (β) of the mirror.

**Figure 12 micromachines-14-00352-f012:**
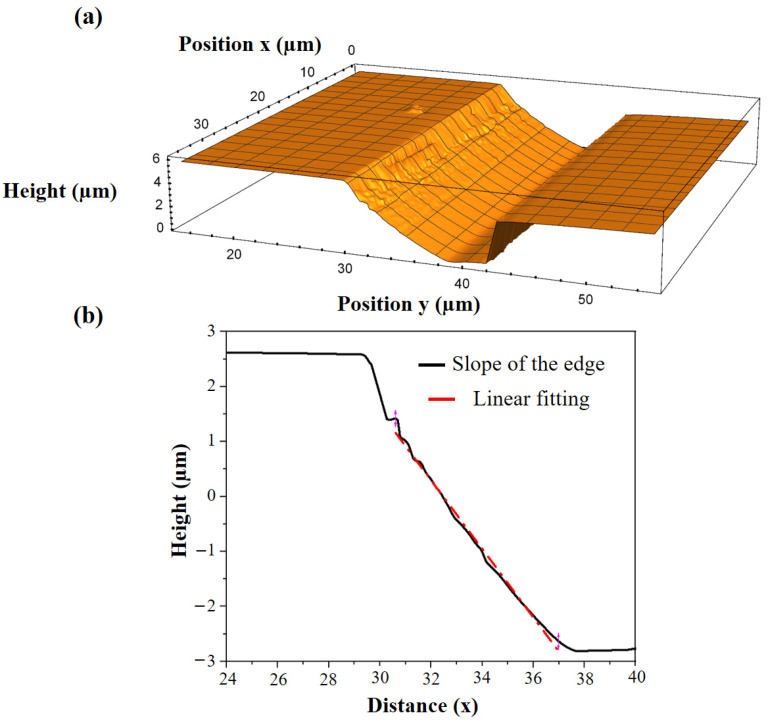
(**a**) 3D optical profile of the emitter edge along with the deflecting mirror and vertical mirror. (**b**) A single scan from this 3D profile along the tilted edge of the proposed 45° deflecting mirror shows the corresponding angle of 29°.

**Figure 13 micromachines-14-00352-f013:**
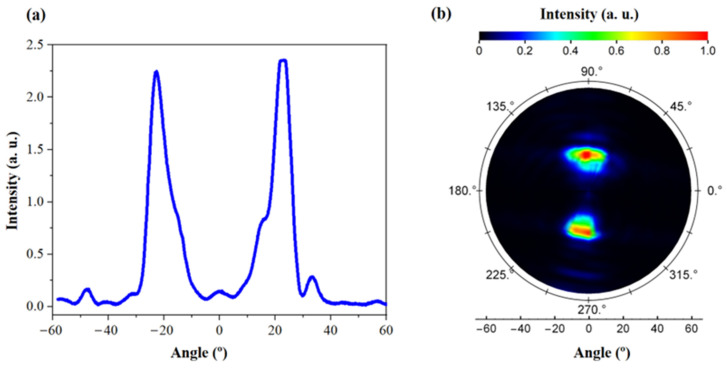
(**a**) Far field intensity distribution of an individual emitter at injection current of 90 mA above laser threshold and (**b**) intensity distribution map at different collection angle of the sensor (photodiode).

**Figure 14 micromachines-14-00352-f014:**
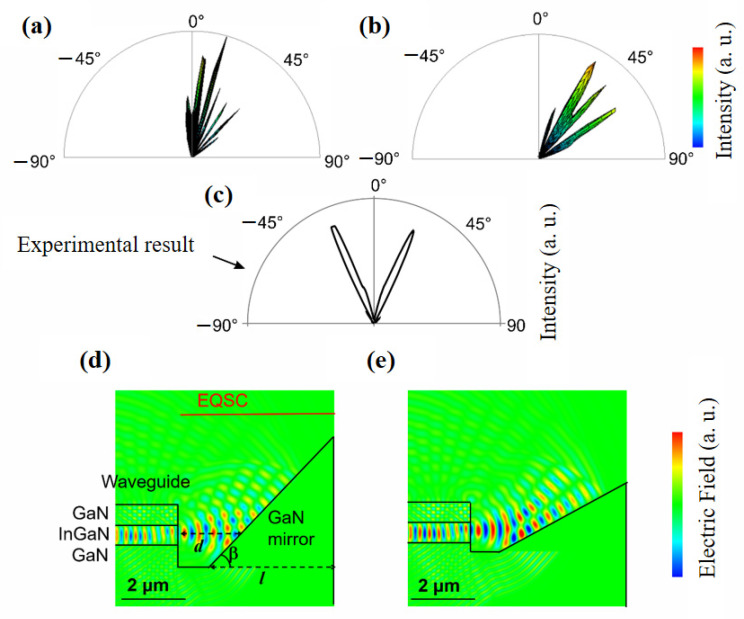
(**a**,**b**) Calculated radiation patterns and (**d**,**e**) electric field distribution for the combination of the InGaN/GaN waveguide and GaN mirror. The cases of mirror angles of b = 45° and 30° are simulated. (**c**) Measured far-field radiation pattern (the same data as shown in Figure 13a) presents in an analogical way to the results of calculations.

## Data Availability

Data is available upon contacting the corresponding author.

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
