# Peer review of "Monolithic 45 Degree Deflecting Mirror as a Key Element for Realization of 2D Arrays of Laser Diodes Based on AlInGaN Semiconductors"

_micromachines, 2023, doi:10.3390/mi14020352_

Round 1

Reviewer 1 Report

The authors present a solution for the realization of 2D arrays of surface emitting GaN-based diode lasers with in-plane laser cavities. To deflect the in-plane emitted laser beams to the out-of-plane direction they make use of microfabricated deflecting mirrors.
Surface emitting lasers are interesting as they allow for easy on-wafer testing, in contrast to edge emitting lasers. Another appealing aspect of surface emitting lasers is that they can easily be fabricated in 2D arrays, which is of interest for optical communication systems and high-power emitters, for example.

While the results presented by the authors are valuable, the manuscript needs significant improvements before it can be published. The authors can find my comments below.

·       * The manuscript contains many typos (e.g. hyphens were they are not required, degree sign missing and used inconsistently, certain Greek letters not displayed in text such as µ and λ, references that seem almost completely wrong). Please fix this.

·       * Line 23: “The technology of deflecting mirrors seems to be a critical one for this application.”
What is ‘this application’ referring to?

·       * Introduction: References 1 to 9 do not seem appropriate, they do not contain the information which is quoted in the manuscript.

·       * Line 37: “complicated and costly pro-cessing and packaging”
Why is processing and packaging complicated and costly for edge emitting lasers? How does this manuscript address this issue?

·       * Line 37: “difficulty in production of laser diode bars (1D solution)”
Why is it difficult to make 1D laser diode bars using edge emitting lasers?

·       * Line 38: What is meant with stacked laser diodes?

·       * Line 39: “The two latter issues are related with thermal chal-lenges and wafer bow characteristic for highly strained InGaAlN laser diode hetero-structures.”
Explain in more detail how these statements are linked with the previous sentences. Is this manuscript addressing these issues? How?

·       * Line 41: “realizing com-pact high-power emitters needed for Digital Light Processing (DLP) projectors, auto-mobile headlights”
Please provide references that show that high-power emitters are needed for these applications.

·       * Line 46: Reference 14 is not about VCSELs in contrast to what the manuscript is suggesting.

·       * Reference 15 is full of typos. Unclear which paper this actually is.

·       * Line 46: “circular beam cross-section facilitating the light 46 coupling to optical fibers [14],[16].”
Beam profiles do not seem circular for the lasers in reference 14.

·       * Line 47: “Also, testing of such individual laser devices does not require wafer dicing or mount-ing and can be tested on-wafer before packaging [6],[17].”
What is the importance of reference 17 in this sentence? How does it support the claims in this sentence?

·       * General comment about introduction: Acknowledge efforts of other researchers to implement on-chip out-of-plane deflecting mirrors. Such mirrors have been demonstrated on multiple material platforms including GaN-based platforms. Here are some examples:
https://ieeexplore.ieee.org/document/6782348
https://ieeexplore.ieee.org/document/6155581
https://opg.optica.org/oe/fulltext.cfm?uri=oe-16-19-15052&id=171948
https://opg.optica.org/oe/fulltext.cfm?uri=oe-27-14-19749&id=414937
https://ieeexplore.ieee.org/document/6363523
https://ieeexplore.ieee.org/document/9097386
https://ieeexplore.ieee.org/document/84489
https://www.spiedigitallibrary.org/conference-proceedings-of-spie/6909/1/A-GaN-based-surface-emitting-laser-with-45degree-inclined-mirror/10.1117/12.762502.short?SSO=1
https://aip.scitation.org/doi/10.1063/1.98763
https://ieeexplore.ieee.org/document/229791
https://aip.scitation.org/doi/10.1063/1.101087
https://ieeexplore.ieee.org/document/4137562

·       * Figure 1: Why is there a distance of 2 µm between the laser facet and mirror?

·       * Line 74: “to align the waveguide with the center of the waveguide.”
This seems to be a typo.

·       * Figure 2: Abbreviation ‘QWs’ is not defined.

·       * Figure 3: What is the meaning of ‘multilevel photolithography’ in step 1?

·       * Line 101: “permits the exposure of photoresist having a significant separation distance from the surface of the used photomask.”
Which distance did you use? How did you choose this distance to obtain 45° angles?

·       * Line 111: “Background level illumination is needed to utilize for patterning the full thickness of the resist after development.”
Unclear what this sentence means.

·       * Line 119: “etching in an Inductively Coupled Plasma Reactive Ion Etching (ICP RIE) system.”
Provide more detailed process parameters.

·       * Line 121: “test samples with sapphire overgrown with 6 and 8 μm of GaN.”
Do these GaN-on-sapphire have the same crystal orientation as the GaN substrates used to make the lasers? Why not test on the actual GaN substrates you will use to make lasers?

·       * Line 133-139: Explain function of all layers. Provide all layer thicknesses and doping concentrations. Why is the In composition in the quantum wells in between 6 and 12%; is it not well defined? What is the composition of the barriers?

·       * Line 142: “We defined the laser cavity stripes to be 900 μm long and the laser ridge to be 2 μm wide.”
What is the height of the ridge? A schematic cross-section of the laser showing the different layers and dimensions would be informative. Is it a single (transverse) mode laser?

·       * Line 151: “As the last step, we used Tetramethylammonium hydroxide (TMAH) solution (27%) which was kept at a constant temperature of 80º and stirred continuously for 20 minutes”
Is this an anisotropic etch? How vertical are the resulting laser facets? Do you mean 80°C?

·       * Figure 6: Close-ups of the laser facets and deflecting mirrors would be a nice addition. Can you use SEM to estimate the angle of the laser facets? How vertical are they?

·       * Line 164: “Preliminary results concluded device efficiency of single horizontal-to-vertical emitter to be as high as 0.6 W/A, usually reaching 0.2 W/A/facet or above for the uncoated facets, and a threshold current around 100 mA.”
How do these numbers compare to state-of-the-art GaN-based lasers?

·       * Line 169: “The measured slope efficiency was seriously underestimated”
Unclear what this sentence means. Does this mean the value of 0.2 W/A/facet is incorrect?

·       * Line 171: “it was not covered by a high-reflective coating which we plan to do in our experiments in the future”
Maybe give an example of a high reflectivity coating that could be used and how this would affect efficiency. What is the expected mirror reflectivity with and without coating?

·       * At what wavelength are the lasers emitting? Measured optical spectra would be a nice addition to the manuscript.

·       * Line 195: Give a reference for the microlenses and their specifications.

·       * Figure 11a: Is the width of the peaks as expected? Is the maximum around 24° expected? How do you explain this outcome knowing the angle of the deflecting mirror to be 29°? How vertical are the laser facets? Does this play a role?

·       * Line 234: Several typos, letters and numbers missing.

·       * Line 235: “The simple waveguide structure composed of only the InGaN waveguide and GaN cladding layers is assumed.”
Why not take the quantum well layers into account?

·       * Line 237: Have you experimentally verified the polarization of your lasers?

·       * Figure 12a and 12b: The sidelobes here seem quite strong. In figure 11 there are barely any sidelobes visible. Can you explain why?

·       * Line 252: Why use letter θ instead of β?

Reviewer 2 Report

The present manuscript gave an experimental result of InGaN device with a mirror for horizontal laser emission. It is interesting and solves a burning issue. However, the novelty of the manuscript should be further enhanced – Many similar structures have been reported, so what’s the significance of this study? There are also many flaws and concerns listed below, by addressing which the manuscript may be in favor of reconsideration for acceptance.

Major issues:

1)     It is recommended to re-organize the manuscript – It is hard to follow the transitions between the concepts in the current manuscript. Please be advised that a thread of the main story can be this: device structure –> mechanism -> simulation/experiment validation – discussion. Also, the context should be further proofread.

2)     In all figures’ captions, there should be an indication saying whether this figure is from an experiment or simulation.

3)     Where’s the result of 2D array emission?

4)     Fig. 7’s should add legends showing which curve is which device.

5)     Fig. 8 should add clear text indicating the device structure, direction, etc. Currently, it’s really hard to read what’s going on.

6)     Fig. 8, it is desired to design a “straight upwards” laser emission, however, according to Fig. 8c, there exists a huge angle of the output beam. Is this still within the desire? Pls comment.

7)     Fig. 10b, show what’s the red line and what’s the black line.

8)     Fig. 11b, show the scale bar.

9)     What’s the difference between Fig. 5b and Fig.10b? Is it okay to be combined them to improve the presentation and organization of the paper?

10)  1)      It is recommended to discuss the possibility of homogenous integration of the proposed device with other InGaN-based power management devices, such as 10.1002/pssr.202100527 and 10.1016/j.spmi.2018.06.045, as one of the novelties of the paper.

Minor issues:

11)  The manuscript needs further proofreading. For instance, see abstract, 450 should be 45 (degrees), etc.

12)  Many dash symbols appear in the wrong positions. See line 34 “qual-ity” for instance.

13)  All images, particularly SEM image’s resolution should be further improved. The scale bar is hard to see.

Round 2

Reviewer 1 Report

The authors dealt with most of the remarks mentioned in the first review report, yet several things can still be improved:

·        In the title you say AlInGaN but in the abstract you use InAlGaN. Please be consistent.

·        In line 28 you use reference 3 in the context of general lighting, but reference 3 is about visible light communication, not general lighting.

·        Line 33: This technology offers high optical power, good beam and spectral quality, controlled lifetime and general maturity of this technology

·        Line 37: difficulty in production of laser diode bars (1D solution) and stacked laser diodes (2D solution)
Please provide references for these statements.

·        In your response letter (and also in the manuscript to some extent) you mention that is difficult to make 1D and 2D laser arrays because of the need for hermetic packaging, wafer bowing (making mounting difficult and distorting beam profile), and thermal challenges. I don't see how this manuscript solves these challenges. Won't you still need hermetic packaging, suffer from wafer bowing and heating with the technology presented in this manuscript? I see how your technology makes on wafer testing possible and the realization of 2D arrays without the complex packaging of stacked laser diodes. This makes your technology very interesting, but I don't see how it solves the other challenges you mention. Please clarify in your manuscript.

·        Line 60: Moreover, other research studies show monolithically mirror-integrated surface emitting laser (SEL) diodes on GaN, as-well-as on silicon platforms
You mention GaN here but in the subsequent text you don’t mention any GaN related references. Please add. Also, not only GaN and silicon have been investigated but also InP.

·        Line 67: microrefelctors -> microreflectors

·        Line 86: we can either decrease the distance between these two mirrors
Explain in the manuscript why you didn’t opt for this approach. You mentioned this briefly in the response letter but not in the manuscript.

·        Line 131: The distance between the photomask and wafer is kept at 4-5 μm
Mention briefly in manuscript how you came to this number.

·        Line 161: After the transferring of the epitaxial layers were grown on the patterned substrate…
Correct language in this sentence.

·        Line 168: Above quantum wells undoped 50 nm was formed
50 nm of which material?

·        Can you add a schematic device cross-section to the manuscript? Similar to the one you added in the response letter, but that one seems incorrect; it does not seem to correspond exactly to the layer stack you mention in the manuscript.

·        Line 177: and the laser ridge to be 2 μm wide
Does this result in single mode operation (i.e. only fundamental guided mode)? Add info to manuscript.

·        Line 214: the number of 8×8 emitters
Shouldn’t this be 4x2 emitters?

·        Line 340: If the optical path difference is equal to mβ, where m is an integer, destructive interference occurs.
Why mβ?

·        Line 355: This open the -> This opens the

·        Line 367: efficiencies approximating with the current state of the art

·        Line 388: 5 References

·        In the response letter you mention: Unfortunately, the simulation was performed assuming a zero distance between the bottom of the deflector and the vertical mirror, while in reality there is around 1μm distance. This is probably the reason for the discrepancy. Also, the surface of the real deflector may show a degree of roughness that leads to a certain degree of averaging of the beams.
Can you redo the simulation with the 1 um distance and if possible with the inclusion of roughness? And add these results to the manuscript?

·        Figure 13 a and b: add scale to axes and/or plot in a way so we can directly compare it to figure 12.

·        Add a brief discussion to the manuscript comparing theory, simulation, and measurement results, in particular about the emission angle and the beam divergence (this was already touched upon in the response letter but is not mentioned in the manuscript).

Author Response

Please see the attachment, thank you!

Reviewer 2 Report

All good.

Author Response

Thank you!